# A neurally plausible model learns successor representations in partially observable environments

**Eszter Vértes    Maneesh Sahani**
Gatsby Computational Neuroscience Unit
University College London
London W1T 4JG, UK.
{eszter,maneesh}@gatsby.ucl.ac.uk

## Abstract

Animals need to devise strategies to maximize returns while interacting with their environment based on incoming noisy sensory observations. Task-relevant states, such as the agent's location within an environment or the presence of a predator, are often not directly observable but must be inferred using available sensory information. Successor representations (SR) have been proposed as a middle-ground between model-based and model-free reinforcement learning strategies, allowing for fast value computation and rapid adaptation to changes in the reward function or goal locations. Indeed, recent studies suggest that features of neural responses are consistent with the SR framework. However, it is not clear how such representations might be learned and computed in partially observed, noisy environments. Here, we introduce a neurally plausible model using *distributional successor features*, which builds on the distributed distributional code for the representation and computation of uncertainty, and which allows for efficient value function computation in partially observed environments via the successor representation. We show that distributional successor features can support reinforcement learning in noisy environments in which direct learning of successful policies is infeasible.

## 1   Introduction

Humans and other animals are able to evaluate long-term consequences of their actions and adapt their behaviour to maximize reward across different environments. This behavioural flexibility is often thought to result from the interaction of two adaptive systems implementing model-based and model-free reinforcement learning (RL).

Model-based learning allows for flexible goal-directed behaviour, acquiring an internal model of the environment which is used to evaluate the consequences of actions. As a result, an agent can rapidly adjust its policy to localized changes in the environment or in reward function. But this flexibility comes at a high computational cost, as optimal actions and value functions depend on expensive simulations in the model. Model-free methods, on the other hand, learn cached values for states and actions, enabling rapid action selection. However, this approach is particularly slow to adapt to changes in the task, as adjusting behaviour even to localized changes, e.g. in the placement of the reward, requires updating cached values at all states in the environment. It has been suggested that the brain makes use both of these complementary approaches, and that they may compete for behavioural control (Daw et al., 2005); indeed, several behavioural studies suggest that subjects implement a hybrid of model-free and model-based strategies (Daw et al., 2011; Gläscher et al., 2010).

Successor representations (SR; Dayan, 1993) augment the internal state used by model-free systems by the expected future occupancy of each world state. SRs can be viewed as a *precompiled* representation of the model under a given policy. Thus, learning based on SRs falls between model-free and model-

based approaches and correspondingly can reproduce a range of behaviours (Russek et al., 2017). Recent studies have argued for evidence consistent with SRs in rodent hippocampal and human behavioural data (Stachenfeld et al., 2017; Momennejad et al., 2017).

Motivated by both theoretical and experimental work arguing that neural RL systems operate over latent states and need to handle state uncertainty (Dayan and Daw, 2008; Gershman, 2018; Starkweather et al., 2017), our work takes the successor framework further by considering partially observable environments. Adopting the framework of distributed distributional coding (Vértes and Sahani, 2018), we show how learnt latent dynamical models of the environment can be naturally integrated with SRs defined over the latent space. We begin with short overviews of reinforcement learning in the partially observed setting (section 2); the SR (section 3); and distributed distributional codes (DDCs) (section 4). In section 5, we describe how using DDCs in the generative and recognition models leads to a particularly simple algorithm for learning latent state dynamics and the associated SR.

## 2 Partially observable Markov decision processes

Markov decision processes (MDP) provide a framework for modelling a wide range of sequential decision-making tasks relevant for reinforcement learning. An MDP is defined by a set of states $\mathcal{S}$ and actions $\mathcal{A}$, a reward function $R : \mathcal{S} \times \mathcal{A} \to \mathbb{R}$, and a probability distribution $\mathcal{T}(s'|s, a)$ that describes the Markovian dynamics of the states conditioned on actions of the agent. For notational convenience we will take the reward function to be independent of action, depending only on state; but the approach we describe is easily extended to the more general case. A partially observable Markov decision process (POMDP) is a generalization of an MDP where the Markovian states $s \in \mathcal{S}$ are not directly observable to the agent. Instead, the agent receives observations ($o \in O$) that depend on the current *latent* state via an observation process $\mathcal{Z}(o|s)$. Formally, a POMDP is a tuple: ($\mathcal{S}$, $\mathcal{A}$, $\mathcal{T}$, $R$, $O$, $\mathcal{Z}$, $\gamma$), comprising the objects defined above and a discount factor $\gamma$. POMDPs can be defined over either discrete or continuous state spaces. Here, we focus on the more general continuous case, although the model we present is applicable to discrete state spaces as well.

## 3 The successor representation

As an agent explores an environment, the states it visits are ordered by the agent's policy and the transition structure of the world. State representations that respect this dynamic ordering are likely to be more efficient for value estimation and may promote more effective generalization. This may not be true of the observed state coordinates. For instance, a barrier in a spatial environment might mean that two states with adjacent physical coordinates are associated with very different values.

Dayan (1993) argued that a natural state space for model-free value estimation is one where distances between states reflect the similarity of future paths given the agent's policy. The successor representation (Dayan, 1993; SR) for state $s_i$ is defined as the expected discounted sum of future occupancies for each state $s_j$, given the current state $s_i$:

$$M^\pi(s_i, s_j) = \mathbb{E}_\pi \Big[ \sum_{k=0}^{\infty} \gamma^k \mathbb{I}[s_{t+k} = s_j] \mid s_t = s_i \Big]. \tag{1}$$

That is, in a discrete state space, the SR is a $N \times N$ matrix where $N$ is the number of states in the environment. The SR depends on the current policy $\pi$ through the expectation in the right hand side of eq. 1, taken with respect to a (possibly stochastic) policy $p^\pi(a_t|s_t)$ and environmental transitions $\mathcal{T}(s_{t+1}|s_t, a_t)$. The SR makes it possible to express the value function in a particularly simple form. Following from eq. 1 and the usual definition of the value function:

$$V^\pi(s_i) = \sum_j M^\pi(s_i, s_j) R(s_j), \tag{2}$$

where $R(s_j)$ is the immediate reward in state $s_j$.

The successor matrix $M^\pi$ can be learned by temporal difference (TD) learning (Sutton, 1988), in much the same way as TD is used to update value functions. In particular, the SR is updated according to a TD error:

$$\delta_t(s_j) = \mathbb{I}[s_t = s_j] + \gamma M^\pi(s_{t+1}, s_j) - M^\pi(s_t, s_j), \tag{3}$$

which reflects errors in *state predictions* rather than rewards, a learning signal typically associated with model based RL.

As shown in eq. 2, the value function can be factorized into the SR—i.e., information about expected future states under the policy—and instantaneous reward in each state[1]. This modularity enables rapid policy evaluation under changing reward conditions: for a fixed policy only the reward function needs to be relearned to evaluate $V^{\pi}(s)$. This contrasts with both model-free and model-based algorithms, which require extensive experience or rely on computationally expensive evaluation, respectively, to recompute the value function.

### 3.1 Successor representation using features

The successor representation can be generalized to continuous states $s \in \mathcal{S}$ by using a set of feature functions $\{\psi_i(s)\}$ defined over $\mathcal{S}$. In this setting, the successor representation (also referred to as the successor feature representation or SF) encodes expected feature values instead of occupancies of individual states:

$$M^{\pi}(s_t, i) = \mathbb{E}_{\pi}\Big[ \sum_{k=0}^{\infty} \gamma^k \psi_i(s_{t+k}) \mid s_t \Big] \tag{4}$$

Assuming that the reward function can be written (or approximated) as a linear function of the features: $R(s) = \boldsymbol{w}_{\text{rew}}^T \boldsymbol{\psi}(s)$ (where the feature values are collected into a vector $\boldsymbol{\psi}(s)$), the value function $V(s_t)$ has a simple form analagous to the discrete case:

$$V^{\pi}(s_t) = \boldsymbol{w}_{\text{rew}}^T M^{\pi}(s_t) \tag{5}$$

For consistency, we can use linear function approximation with the same set of features as in eq. 4 to parametrize the successor features $M^{\pi}(s_t, i)$.

$$M^{\pi}(s_t, i) \approx \sum_j U_{ij} \psi_j(s_t) \tag{6}$$

The form of the SFs, embodied by the weights $U_{ij}$, can be found by temporal difference learning:

$$\Delta U_{ij} = \delta_i \psi_j(s_t) \qquad\qquad \delta_i = \psi_i(s_t) + \gamma M(s_{t+1}, i) - M(s_t, i) \tag{7}$$

As we have seen in the discrete case, the TD error here signals prediction errors about features of state, rather than about reward.

## 4 Distributed distributional codes

Distributed distributional codes (DDC) are a candidate for the neural representation of uncertainty (Zemel et al., 1998; Sahani and Dayan, 2003) and recently have been shown to support accurate inference and learning in hierarchical latent variable models (Vértes and Sahani, 2018). In a DDC, a population of neurons represent distributions in their firing rates implicitly, as a set of expectations:

$$\boldsymbol{\mu} = \mathbb{E}_{p(s)}[\boldsymbol{\psi}(s)] \tag{8}$$

where $\boldsymbol{\mu}$ is a vector of firing rates, $p(s)$ is the represented distribution, and $\boldsymbol{\psi}(s)$ is a vector of encoding functions specific to each neuron. DDCs can be thought of as representing exponential family distributions with sufficient statistics $\boldsymbol{\psi}(s)$ using their mean parameters $\mathbb{E}_{p(s)}[\boldsymbol{\psi}(s)]$ (Wainwright and Jordan, 2008).

## 5 Distributional successor representation

As discussed above, the successor representation can support efficient value computation by incorporating information about the policy and the environment into the state representation. However, in more realistic settings, the states themselves are not directly observable and the agent is limited to state-dependent noisy sensory information.

**Algorithm 1** Wake-sleep algorithm in the DDC state-space model
---
Initialise $T, W$
**while** not converged **do**
   **Sleep phase:**
   sample: $\{s_t^{\text{sleep}}, o_t^{\text{sleep}}\}_{t=0\ldots N} \sim p(\mathcal{S}_N, \mathcal{O}_N)$
   update $W$: $\Delta W \propto \sum\limits_t \left( \boldsymbol{\psi}(s_t^{\text{sleep}}) - f_W(\boldsymbol{\mu}_{t-1}(\mathcal{O}_{t-1}^{\text{sleep}}), o_t^{\text{sleep}}) \right) \nabla_W f_W$

   **Wake phase:**
   $\mathcal{O}_N \leftarrow \{\text{collect observations}\}$
   infer posterior DDC $\boldsymbol{\mu}_t(\mathcal{O}_t) = f_W(\boldsymbol{\mu}_{t-1}(\mathcal{O}_{t-1}), o_t)$
   update $T$: $\Delta T \propto (\boldsymbol{\mu}_{t+1}(\mathcal{O}_{t+1}) - T\boldsymbol{\mu}_t(\mathcal{O}_t))\boldsymbol{\mu}_t(\mathcal{O}_t)^T$
   update observation model parameters
**end while**
---

In this section, we lay out how the DDC representation for uncertainty allows for learning and computing with successor representations defined over latent variables. First, we describe an algorithm for learning and inference in dynamical latent variable models using DDCs. We then establish a link between the DDC and successor features (eq. 4) and show how they can be combined to learn what we call the *distributional successor features*. We discuss different algorithmic and implementation-related choices for the proposed scheme and their implications.

### 5.1 Learning and inference in a state space model using DDCs

Here, we consider POMDPs where the state-space transition model is itself defined by a conditional DDC with means that depend linearly on the preceding state features. That is, the conditional distribution describing the latent dynamics implied by following the policy $\pi$ can be written in the following form:

$$p^\pi(s_{t+1}|s_t) \Leftrightarrow \mathbb{E}_{s_{t+1}|s_t,\pi}[\boldsymbol{\psi}(s_{t+1})] = T^\pi \boldsymbol{\psi}(s_t) \qquad (9)$$

where $T^\pi$ is a matrix parametrizing the functional relationship between $s_t$ and the expectation of $\boldsymbol{\psi}(s_{t+1})$ with respect to $p^\pi(s_{t+1}|s_t)$.

The agent has access only to sensory observations $o_t$ at each time step, and in order to be able to make use of the underlying latent structure, it has to learn the parameters of generative model $p(s_{t+1}|s_t)$, $p(o_t|s_t)$ as well as learn to perform inference in that model.

We consider online inference (filtering), i.e. at each time step $t$ the recognition model produces an estimate $q(s_t|\mathcal{O}_t)$ of the posterior distribution $p(s_t|\mathcal{O}_t)$ given all observations up to time $t$: $\mathcal{O}_t = (o_1, o_2, \ldots o_t)$. As in the DDC Helmholtz machine (Vértes and Sahani, 2018), these distributions are represented by a set of expectations—i.e., by a DDC:

$$\boldsymbol{\mu}_t(\mathcal{O}_t) = \mathbb{E}_{q(s_t|\mathcal{O}_t)}[\boldsymbol{\psi}(s_t)] \qquad (10)$$

The filtering posterior $\boldsymbol{\mu}_t(\mathcal{O}_t)$ is computed iteratively, using the posterior in the previous time step $\boldsymbol{\mu}_{t-1}(\mathcal{O}_{t-1})$ and the new observation $o_t$. The Markovian structure of the state space model (see fig. 1) ensures that the recognition model can be written as a recursive function:

$$\boldsymbol{\mu}_t(\mathcal{O}_t) = f_W(\boldsymbol{\mu}_{t-1}(\mathcal{O}_{t-1}), o_t) \qquad (11)$$

with a set of parameters $W$.

The recognition and generative models are updated using an adapted version of the wake-sleep algorithm (Hinton et al., 1995; Vértes and Sahani, 2018). In the following, we describe the two phases of the algorithm in more detail (see Algorithm 1).

**Sleep phase**

The aim of the sleep phase is to adjust the parameters of the recognition model given the current generative model. Specifically, the recognition model should approximate the expectation of the DDC encoding functions $\boldsymbol{\psi}(s_t)$ under the filtering posterior $p(s_t|\mathcal{O}_t)$. This can be achieved by moment matching, i.e., simulating a sequence of latent and observed states from the current model and

minimizing the Euclidean distance between the output of the recognition model and the sufficient statistic vector $\boldsymbol{\psi}(.)$ evaluated at the latent state from the next time step.

$$W \leftarrow \underset{W}{\operatorname{argmin}} \sum_t \|\boldsymbol{\psi}(s_t^{\text{sleep}}) - f_W(\boldsymbol{\mu}_{t-1}(\mathcal{O}_{t-1}^{\text{sleep}}), o_t^{\text{sleep}})\|^2 \tag{12}$$

where $\{s_t^{\text{sleep}}, o_t^{\text{sleep}}\}_{t=0...N} \sim p(s_0)p(o_0|s_0) \prod_{t=0}^{N-1} p(s_{t+1}|s_t, T^\pi)p(o_{t+1}|s_{t+1})$.

This update rule can be implemented online as samples are simulated, and after a sufficiently long simulated sequence (or multiple sequences) $\{s_t^{\text{sleep}}, o_t^{\text{sleep}}\}_t$ the recognition model will learn to approximate expectations of the form: $f_W(\boldsymbol{\mu}_{t-1}(\mathcal{O}_{t-1}^{\text{sleep}}), o_t^{\text{sleep}}) \approx \mathbb{E}_{p(s_t|\mathcal{O}_t)}[\boldsymbol{\psi}(s_t)]$, yielding a DDC representation of the posterior.

**Wake phase**

In the wake phase, the parameters of the generative model are adapted such that it captures the sensory observations better. Here, we focus on learning the policy-dependent latent dynamics $p^\pi(s_{t+1}|s_t)$; the observation model can be learned by the approach of Vértes and Sahani (2018). Given a sequence of inferred posterior representations $\{\boldsymbol{\mu}_t(\mathcal{O}_t)\}$ computed using wake phase observations, the parameters of the latent dynamics $T$ can be updated by minimizing a simple predictive cost function:

$$T \leftarrow \underset{T}{\operatorname{argmin}} \sum_t \|\boldsymbol{\mu}_{t+1}(\mathcal{O}_{t+1}) - T\boldsymbol{\mu}_t(\mathcal{O}_t)\|^2 \tag{13}$$

The intuition behind eq. 13 is that for the optimal generative model the latent dynamics satisfies the following equality: $T^*\boldsymbol{\mu}_t(\mathcal{O}_t) = \mathbb{E}_{p(o_{t+1}|\mathcal{O}_t)}[\boldsymbol{\mu}_{t+1}(\mathcal{O}_{t+1})]$. That is, the predictions made by combining the posterior at time $t$ and the prior will agree with the average posterior at the next time step—making $T^*$ a stationary point of the optimization in eq. 13. For further details on the nature of the approximation implied by the wake phase update and its relationship to variational learning, see the supplementary material. In practice, the update can be done online, using gradient steps analogous to prediction errors:

$$\Delta T \propto (\boldsymbol{\mu}_{t+1}(\mathcal{O}_{t+1}) - T\boldsymbol{\mu}_t(\mathcal{O}_t))\boldsymbol{\mu}_t(\mathcal{O}_t)^T \tag{14}$$

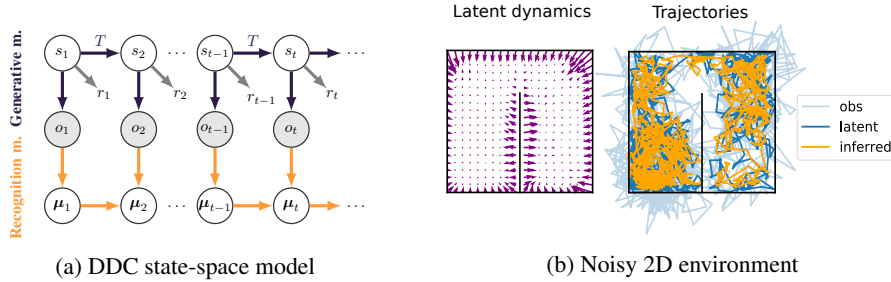

(a) DDC state-space model          (b) Noisy 2D environment

Figure 1: Learning and inference in a state-space model parametrized by a DDC. (a) The structure of the generative and recognition models. (b) Visualization of the dynamics $T$ learned by the wake-sleep (algorithm 1). Arrows show the conditional mean $\mathbb{E}_{s_{t+1}|s_t}[s_{t+1}]$ for each location. (c) Posterior mean trajectories inferred using the recognition model, plotted on top of true latent and observed trajectories.

Figure 1 shows a state-space model corresponding to a random walk policy in the latent space with noisy observations, learned using DDCs (Algorithm 1). For further details of the experiment, see the supplementary material.

## 5.2 Learning distributional successor features

Next, we show how using a DDC to parametrize the generative model (eq. 9) makes it possible to compute the successor features defined in the latent space in a tractable form, and how this computation can be combined with inference based on sensory observations.

Following the definition of the SFs (eq. 4), we have:

$$M(s_t) = \mathbb{E}_\pi \Big[ \sum_{k=0}^\infty \gamma^k \boldsymbol{\psi}(s_{t+k})|s_t \Big] = \sum_{k=0}^\infty \gamma^k \mathbb{E}_\pi [\boldsymbol{\psi}(s_{t+k})|s_t] \tag{15}$$

We can compute the conditional expectations of the feature vector $\boldsymbol{\psi}$ in eq. 15 by applying the dynamics $k$ times to the features $\boldsymbol{\psi}(s_t)$: $\mathbb{E}_{s_{t+k}|s_t}[\boldsymbol{\psi}(s_{t+k})] = T^k \boldsymbol{\psi}(s_t)$. Thus, we have:

$$M(s_t) = \sum_{k=0}^\infty \gamma^k T^k \boldsymbol{\psi}(s_t) \tag{16}$$

$$= (I - \gamma T)^{-1} \boldsymbol{\psi}(s_t) \tag{17}$$

Eq. 17 is reminiscent of the result for discrete observed state spaces $M(s_i, s_j) = (I - \gamma P)_{ij}^{-1}$ (Dayan, 1993), where P is a matrix containing Markovian transition probabilities between states. In a continuous state space, however, finding a closed form solution like eq. 17 is non-trivial, as it requires evaluating a set of typically intractable integrals. The solution presented here directly exploits the DDC parametrization of the generative model and the correspondence between the features used in the DDC and the SFs.

In this framework, we can not only compute the successor features in closed form in the latent space, but also evaluate the *distributional successor features*, the posterior expectation of the SFs given a sequence of sensory observations:

$$\mathbb{E}_{s_t|\mathcal{O}_t}[M(s_t)] = (I - \gamma T)^{-1} \mathbb{E}_{s_t|\mathcal{O}_t}[\boldsymbol{\psi}(s_t)] \tag{18}$$

$$= (I - \gamma T)^{-1} \boldsymbol{\mu}_t(\mathcal{O}_t) \tag{19}$$

The results from this section suggest a number of different ways the distributional successor features $\mathbb{E}_{s_t|\mathcal{O}_t}[M(s_t)]$ can be learned or computed.

### 5.2.1 Learning distributional SFs during *sleep phase*

The matrix $U = (I - \gamma T)^{-1}$ needed to compute distributional SFs in eq. 19 can be learned from temporal differences in feature predictions based on *sleep phase* simulated latent state sequences (see eq. 6-7). Following a potential change in the dynamics of the environment, sleep phase learning allows for updating SFs and therefore cached values offline, without the need for further experience.

### 5.2.2 Computing distributional SFs by *dynamics*

Alternatively, eq. 19 can be implemented as a fixed point of a linear dynamical system, with recurrent connections reflecting the model of the latent dynamics:

$$\tau \dot{x} = -x + \gamma T x + \boldsymbol{\mu}_t(\mathcal{O}_t) \tag{20}$$

$$\Rightarrow x(\infty) = (I - \gamma T)^{-1} \boldsymbol{\mu}_t(\mathcal{O}_t) \tag{21}$$

In this case, there is no need to learn $(I - \gamma T)^{-1}$ explicitly but it is implicitly computed through dynamics. For this to work, there is an underlying assumption that the dynamical system in eq. 20 reaches equilibrium on a timescale ($\tau$) faster than that on which the observations $\mathcal{O}_t$ evolve.

Both of these approaches avoid having to compute the matrix inverse directly and allow for evaluation of policies given by a corresponding dynamics matrix $T^\pi$ offline.

### 5.2.3 Learning distributional SFs during *wake phase*

Instead of fully relying on the learned latent dynamics to compute the distributional SFs, we can use posteriors computed by the recognition model during the wake phase, that is, using observed data. We can define the distributional SFs directly on the DDC posteriors: $\widetilde{M}(\mathcal{O}_t) = \mathbb{E}_\pi[\sum_k \gamma^k \boldsymbol{\mu}_{t+k}(\mathcal{O}_{t+k})|\boldsymbol{\mu}_t(\mathcal{O}_t)]$, treating the posterior representation $\boldsymbol{\mu}_t(\mathcal{O}_t)$ as a feature space over sequences of observations $\mathcal{O}_t = (o_1 \ldots o_t)$. Analogously to section 3.1, $\widetilde{M}(\mathcal{O}_t)$ can be acquired by

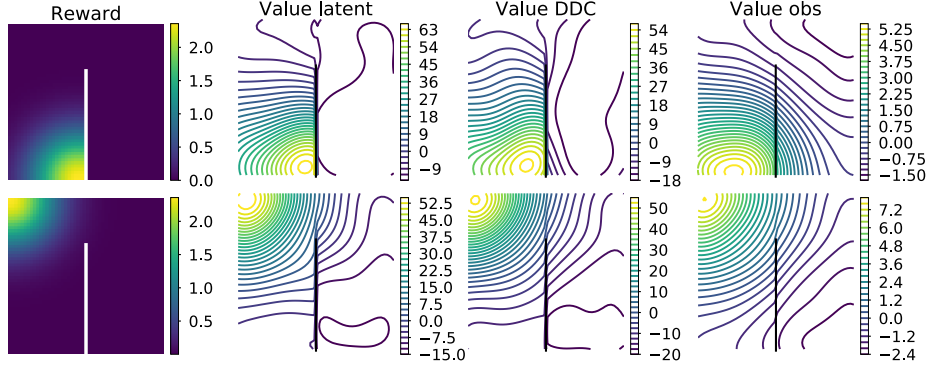

Figure 2: Value functions under a random walk policy for two different reward locations. Values were computed using SFs based on the latent, inferred DDC posterior or observed state variables.

TD learning and assuming linear function approximation: $\widetilde{M}(\mathcal{O}_t) \approx U\boldsymbol{\mu}_t(\mathcal{O}_t)$. The matrix $U$ can be updated online, while executing a given policy and continuously inferring latent state representations using the recognition model:

$$\Delta U \propto \delta_t \boldsymbol{\mu}_t(\mathcal{O}_t)^T \tag{22}$$
$$\delta_t = \boldsymbol{\mu}_t(\mathcal{O}_t) + \gamma M(\mathcal{O}_{t+1}) - M(\mathcal{O}_t) \tag{23}$$

It can be shown that $\widetilde{M}(\mathcal{O}_t)$, as defined here, is equivalent to $\mathbb{E}_{s_t|\mathcal{O}_t}[M(s_t)]$ if the learned generative model is optimal–assuming no model mismatch–and the recognition model correctly infers the corresponding posteriors $\boldsymbol{\mu}_t(\mathcal{O}_t)$ (see supplementary material). In general, however, exchanging the order of TD learning and inference leads to different SFs. The advantage of learning the distributional successor features in the wake phase is that even when the model does not perfectly capture the data (e.g. due to lack of flexibility or early on in learning) the learned SFs will reflect the structure in the observations through the posteriors $\boldsymbol{\mu}_t(\mathcal{O}_t)$.

## 5.3 Value computation in a noisy 2D environment

We illustrate the importance of being able to consistently handle uncertainty in the SFs by learning value functions in a noisy environment. We use a simple 2-dimensional box environment with continuous state space that includes an internal wall. The agent does not have direct access to its spatial coordinates, but receives observations corrupted by Gaussian noise. Figure 2 shows the value functions computed using the successor features learned in three different settings: assuming direct access to latent states, treating observations as though they were noise-free state measurements, and using latent state estimates inferred from observations. The value functions computed in the latent space and computed from DDC posterior representations both reflect the structure of the environment, while the value function relying on SFs over the observed states fails to learn about the barrier.

To demonstrate that this is not simply due to using the suboptimal random walk policy, but persists through learning, we have learned successor features while adjusting the policy to a given reward function (see figure 3). The policy was learned by generalized policy iteration (Sutton and Barto, 1998), alternating between taking actions following a greedy policy and updating the successor features to estimate the corresponding value function.

The value of each state and action was computed from the value function $V(s)$ by a one-step look-ahead, combining the immediate reward with the expected value function having taken a given action:

$$Q(s_t, a_t) = r(s_t) + \gamma \mathbb{E}_{s_{t+1}|s_t, a_t}[V(s_{t+1})] \tag{24}$$

In our case, as the value function in the latent space is expressed as a linear function of the features $\boldsymbol{\psi}(s)$: $V(s) = \boldsymbol{w}_{\text{rew}}^T U \boldsymbol{\psi}(s)$ (eq. 5-6), the expectation in 24 can be expressed as:

$$\mathbb{E}_{s_{t+1}|s_t, a_t}[V(s_{t+1})] = \boldsymbol{w}_{\text{rew}}^T U \cdot \mathbb{E}_{s'|s, a}[\boldsymbol{\psi}(s_{t+1})] \tag{25}$$
$$\approx \boldsymbol{w}_{\text{rew}}^T U \cdot P \cdot (\boldsymbol{\psi}(s_t) \otimes \boldsymbol{\phi}(a_t)) \tag{26}$$

Where $P$ is a linear mapping, $P : \Psi \times \Phi \to \Psi$, that contains information about the distribution $p(s_{t+1}|s_t, a_t)$. More specifically, $P$ is trained to predict $\mathbb{E}_{s_{t+1}|s_t,a_t}[\boldsymbol{\psi}(s_{t+1})]$ as a bilinear function of state and action features $(\boldsymbol{\psi}(s_t), \boldsymbol{\phi}(a_t))$. Given the state-action value, we can implement a greedy policy by choosing actions that maximize $Q(s, a)$:

$$a^* = \underset{a \in \mathcal{A}}{\operatorname{argmax}} \, Q(s_t, a_t) \tag{27}$$

$$= \underset{a \in \mathcal{A}}{\operatorname{argmax}} \, r(s_t) + \gamma \boldsymbol{w}_{\text{rew}}^T U \cdot P \cdot (\boldsymbol{\psi}(s_t) \times \boldsymbol{\phi}(a_t)) \tag{28}$$

The argmax operation in eq. 28 (possibly over a continuous space of actions) could be biologically implemented by a ring attractor where the neurons receive state-dependent input through feedforward weights reflecting the tuning $(\phi(a))$ of each neuron in the ring.

Just as in figure 2, we compute the value function in the fully observed case, using inferred states or using only the noisy observations. For the latter two, we replace $\boldsymbol{\psi}(s_t)$ in eq. 28 with the inferred state representation $\boldsymbol{\mu}(\mathcal{O}_t)$ and the observed features $\boldsymbol{\psi}(o_t)$, respectively. As the agent follows the greedy policy and it receives new observations the corresponding SFs are adapted accordingly. Figure 3 shows the learned value functions $V^\pi(s)$, $V^\pi(\boldsymbol{\mu})$ and $V^\pi(o)$ for a given reward location and the corresponding dynamics $T^\pi$. The agent having access to the true latent state as well as the one using distributional SFs successfully learn policies leading to the rewarded location. As before, the agent learning SFs purely based on observations remains highly sub-optimal.

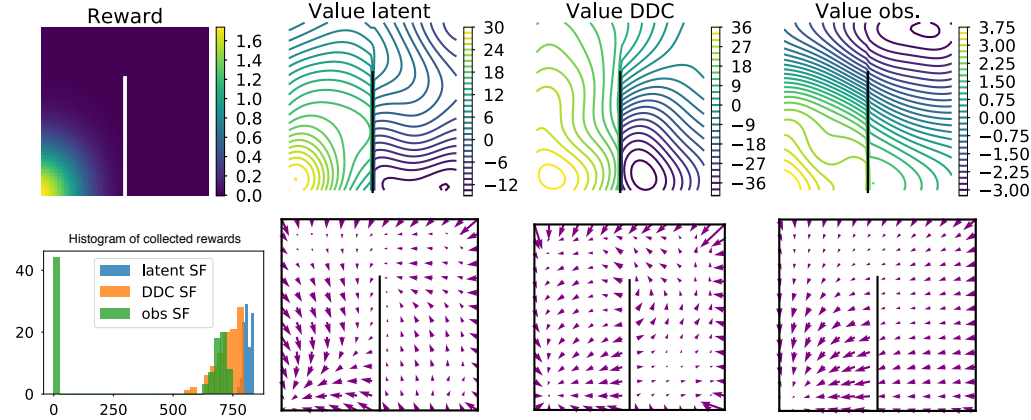

Figure 3: Value functions computed by SFs under the learned policy. Top row shows reward and value functions learned in the three different conditions. Bottom row shows histogram of collected rewards from 100 episodes with random initial states, and the learned dynamics $T^\pi$ visualized as in fig. 1.

# 6 Discussion and related work

We have shown that the DDC represention of uncertainty over latent variables can be naturally integrated with representations of uncertainty about future states, and thus offers a natural generalisation of SRs to more realistic environments with partial observability. The proposed algorithm jointly tackles the problem of learning the latent variable model and learning to perform online inference by filtering. Distributional SFs are applicable to POMDPs with continuous or discrete variables and leverage a flexible posterior approximation, not restricted to a simple parametric form, that is represented in a population of neurons in a distributed fashion.

While parametrising the latent dynamics with DDCs is attractive as it makes computing the SFs in the latent space analytically tractable and allows for computing distributional SFs by recurrent dynamics (sec. 5.2.2), it is so far unclear how sampling from such a model might be implemented by neural circuits. Alternatively, one can consider a standard exponential family parametrisation which remains compatible with sleep and wake phase TD learning of distributional SFs.

Earlier work on biological reinforcement learning in POMDPs was restricted to the case of binary or categorical latent variables where the posterior beliefs can be computed analytically (Rao, 2010).

Furthermore the transition model of the POMDP was assumed to be known, rather than learned as in the present work.

Here, we have defined distributional SFs over states, using single step look-ahead to compute state-action values (eq. 24). Alternatively, SFs could be defined directly over both states and actions (Kulkarni et al., 2016; Barreto et al., 2017) whilst retaining the distributional development presented here. Barreto et al. (2017, 2019) have shown that successor representations corresponding to previously learned tasks can be used as a basis to construct policies for novel tasks, enabling generalization. Our framework can be extended in a similar way, eliminating the need to adapt the SFs as the policy of the agent changes.

The neurotransmitter dopamine has long been hypothesised to signal reward prediction errors (RPE) and thus to play a key role in temporal difference learning (Schultz et al., 1997). More recently, it has been argued that dopamine activity is consistent with RPEs computed based on belief states rather than sensory observations directly (Babayan et al., 2018; Lak et al., 2017; Sarno et al., 2017). Thus dopamine is well suited to carry the information necessary for learning value functions under state uncertainty. In another line of experimental work, dopamine has been found to signal *sensory* prediction errors (PE) even if the absence of an associated change in value (Takahashi et al., 2017), suggesting a more general role of dopamine in learning (Gershman, 2018; Gardner et al., 2018). Gardner et al. have proposed that dopamine—signalling prediction error over features of state—may provide the neural substrate for the error signals necessary to learn successor representations. Distributional SFs unify these two sets of observations and their theoretical implications in a single framework. They posit that PEs are computed over the posterior belief about latent states (represented as DDCs), and that these PEs are defined over a set of non-linear features of the hidden state rather than reward.

The proposed learning scheme for distributional SFs allows for flexible interpolation between model-based and model-free approaches. Wake phase learning of SFs is grounded in observations and only relies on the model through the belief state updates, while sleep phase learning uses simulated latent states from the model to update the SFs—akin to the Dyna algorithm (Sutton, 1990).

The framework for learning distributional successor features presented here also provides a link between various intriguing and seemingly disparate experimental observations in the hippocampus. The relationship between hippocampal place cell activity and (non-distributional) SRs has been explored previously (e.g., Stachenfeld et al., 2014; 2017) providing an interpretation for phenomena such as "splitter" cells, which show spatial tuning that depends on the whole trajectory (i.e. policy) traversed by the animal not just on its current position (Grieves et al., 2016). However, as discussed earlier, relevant states for a given reinforcement learning problem (in this case states over which the SR should be learned) cannot be assumed to be directly available to the agent but must be inferred from observations. The hypothesis that hippocampal place cell activity encodes *inferred* location, with its concomitant uncertainty, has also been linked to experimental data (Madl et al., 2014). Thus, our approach connects these two separate threads in the literature and thereby encompasses both groups of experimental results.

Lastly, the framework helps to link simulation of an internal model to learning. Acquisition of the inference model in our framework requires simulating experience (sleep samples) from the agent's current model of the environment, to provide the basis for an update of the recognition model. The sleep samples reflect the agent's knowledge of the environmental dynamics but they need not correspond exactly to a previously experienced trajectory. This is reminiscent of hippocampal "replay" which does not just recapitulate previous experience, but often represents novel trajectories not previously experienced by the animal (Gupta et al., 2010; Ólafsdóttir et al., 2015; Stella et al., 2019). Relatedly, Liu et al. (2019) recently observed that replay events in humans reflect abstract structural knowledge of a learned task. Our model suggests a novel functional interpretation of these replayed trajectories; namely, that they may play an important role in *learning to infer* relevant latent states from observations. This accords with the observation that experimental interference with replay events impedes learning in contexts where optimal actions depend on history-based inference (Jadhav et al., 2012).

Distributional SFs provide interpretation for a variety of experimental observations and a step towards algorithmic solutions for flexible decision making in realistic and challenging problem settings animals face, i.e. under state uncertainty.

## Footnotes

[1]Alternatively, for the more general case of action-dependent reward, the expected instantaneous reward under the policy-dependent action in each state.

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
