[Supplementary Material]

# Supplementary material

## A Approximations in the wake phase update

Here, we give some additional insights into the nature of the approximation implied by the wake phase update for the DDC state-space model and discuss its link to variational methods.

According to the standard M step in variational EM, the model parameters are updated to maximize the expected log-joint of the model under the approximate posterior distributions:

$$\Delta\theta \propto \nabla_\theta \sum_t \mathbb{E}_{q(s_t, s_{t+1}|\mathcal{O}_{t+1})}[\log p_\theta(s_{t+1}|s_t)] \tag{1}$$

$$= \nabla_\theta \sum_t - \int q(s_t, s_{t+1}|\mathcal{O}_{t+1})(\log p_\theta(s_{t+1}|s_t) + \log q(s_t|\mathcal{O}_{t+1}))d(s_t, s_{t+1}) \tag{2}$$

$$= \nabla_\theta \sum_t -KL[q(s_t, s_{t+1}|\mathcal{O}_{t+1})\|p_\theta(s_{t+1}|s_t)q(s_t|\mathcal{O}_{t+1})] \tag{3}$$

After projecting the distributions appearing in the KL divergence (eq. 3) into the joint exponential family defined by sufficient statistics $[\psi(s_t), \psi(s_{t+1})]$, they can be represented using the corresponding mean parameters:

$$q(s_t, s_{t+1}|\mathcal{O}_{t+1}) \xrightarrow{\mathcal{P}} \left[ \begin{array}{c} \mathbb{E}_{q(s_t, s_{t+1}|\mathcal{O}_{t+1})}[\psi(s_t)] \\ \mathbb{E}_{q(s_t, s_{t+1}|\mathcal{O}_{t+1})}[\psi(s_{t+1})] \end{array} \right] = \left[ \begin{array}{c} \mu_t(\mathcal{O}_{t+1}) \\ \mu_{t+1}(\mathcal{O}_{t+1}) \end{array} \right] \tag{4}$$

$$p_\theta(s_{t+1}|s_t)q(s_t|\mathcal{O}_{t+1}) \xrightarrow{\mathcal{P}} \left[ \begin{array}{c} \mathbb{E}_{p_\theta(s_{t+1}|s_t)q(s_t|\mathcal{O}_{t+1})}[\psi(s_t)] \\ \mathbb{E}_{p_\theta(s_{t+1}|s_t)q(s_t|\mathcal{O}_{t+1})}[\psi(s_{t+1})] \end{array} \right] = \left[ \begin{array}{c} \mu_t(\mathcal{O}_{t+1}) \\ T\mu_t(\mathcal{O}_{t+1}) \end{array} \right] \tag{5}$$

To restrict ourselves to online inference, we can make a further approximation: $\mu_t(\mathcal{O}_{t+1}) \approx \mu_t(\mathcal{O}_t)$. Thus, the wake phase update can be thought of as replacing the KL divergence in equation 3 by the Euclidean distance between the (projected) mean parameter representations in eq. 4-5.

$$\sum_t \|\mu_{t+1}(\mathcal{O}_{t+1}) - T\mu_t(\mathcal{O}_t)\|^2 \tag{6}$$

Note that this cost function is directly related to the maximum mean discrepancy (Gretton et al. [2012]; MMD)–a non-parametric distance metric between two distributions–with a finite dimensional RKHS.

## B Equivalence of sleep and wake phase TD

Here, we show that the posterior expectation of the SFs learned in the latent space during sleep phase ($\mathbb{E}_{p(s_t|\mathcal{O}_t)}[M(s_t)]$) is equivalent to the SFs learned during wake phase ($\widetilde{M}(\mu_t(\mathcal{O}_t))$) if the generative model matches the data distribution and the recognition model produces the exact posterior.

Wake phase TD learns to approximate the following expression:

$$\widetilde{M}(\mu_t(\mathcal{O}_t)) = \mathbb{E}_{p(\mathcal{O}_{>t}|\mathcal{O}_t)}[\sum_{k=0}^{\infty} \gamma^k \mu_{t+k}(\mathcal{O}_{t+k})] \tag{7}$$

Where

$$\mathbb{E}_{p(\mathcal{O}_{>t}|\mathcal{O}_t)}[\mu_{t+k}(\mathcal{O}_{t+k})] = \mathbb{E}_{p(\mathcal{O}_{t+1:t+k}|\mathcal{O}_t)}[\mu_{t+k}(\mathcal{O}_{t+k})] \tag{8}$$

$$= \int d\mathcal{O}_{t+1:t+k}\ p(\mathcal{O}_{t+1:t+k}|\mathcal{O}_t) \int ds_{t+k}\ p(s_{t+k}|\mathcal{O}_{t+k})\psi(s_{t+k})$$

$$= \int d\mathcal{O}_{t+1:t+k}\ p(\mathcal{O}_{t+1:t+k}|\mathcal{O}_t) \int ds_{t+k}\ \frac{p(s_{t+k},\mathcal{O}_{t+1:t+k}|\mathcal{O}_t)}{p(\mathcal{O}_{t+1:t+k}|\mathcal{O}_t)}\psi(s_{t+k})$$

$$= \int ds_{t+k} \int d\mathcal{O}_{t+1:t+k}\ p(s_{t+k},\mathcal{O}_{t+1:t+k}|\mathcal{O}_t)\psi(s_{t+k})$$

$$= \int ds_{t+k} p(s_{t+k}|\mathcal{O}_t)\psi(s_{t+k}) \tag{9}$$

$$= T^k \mu_t$$

Thus, we have:

$$\widetilde{M}(\mu_t(\mathcal{O}_t)) = \sum_{k=0}^{\infty} \gamma^k T^k \mu_t \tag{10}$$

$$= (I - \gamma T)^{-1}\mu_t$$

$$= \mathbb{E}_{p(s_t|\mathcal{O}_t)}[M(s_t)]$$

# C Further experimental details

**Figure 1: Learning and inference in the DDC state-space model**
The generative model corresponding to a random walk policy:

$$p(s_{t+1}|s_t) = [s_t + \tilde{\eta}]_{\text{WALLS}}, \tag{11}$$
$$p(o_t|s_t) = s_t + \xi$$

Where $[.]_{\text{WALLS}}$ indicates the constraints introduced by the walls in the environment (outer walls are of unit length). $\eta \sim \mathcal{N}(0, \sigma_s = 1.)$, $\tilde{\eta} = 0.06 * \eta/\|\eta\|$, $\xi \sim \mathcal{N}(0, \sigma_o = 0.1)$, $s_t, o_t \in \mathbb{R}^2$
We used K=100 Gaussian features with width $\sigma_\psi = 0.3$ for both the latent and observed states. A small subset of features were truncated along the internal wall, to limit the artifacts from the function approximation. Alternatively, a features with various spatial scales could also be used. The recursive recognition model was parametrized linearly using the features:

$$f_W(\mu_{t-1}, o_t) = W[T\mu_{t-1}; \psi(o_t)] \tag{12}$$

As sampling from the DDC parametrized latent dynamics is not tractable in general, in the sleep phase, we generated approximate samples from a Gaussian distribution with consistent mean.
The generative and recognition models were trained through 50 wake-sleep cycles, with $3 \cdot 10^4$ sleep samples, and $5 \cdot 10^4$ wake phase observations.
The latent dynamics in Fig.1b is visualized by approximating the mean dynamics as a linear readout from the DDC: $\mathbb{E}_{s_{t+1}|s_t}[s_{t+1}] \approx \alpha T\psi(s_t)$ where $s \approx \alpha\psi(s)$.

**Figure 2 Values under a random walk policy** To compute the value functions under the random walk policy we computed the SFs based on the latent ($\psi(s)$), inferred ($\mu$) or observed ($\psi(o)$) features, with discount factor $\gamma = 0.99$. In each case, we estimated the reward vector $w_{\text{rew}}$ using the available state information.

**Figure 3 Values under a learned policy** To construct the state-action value function, we used 10 features over actions $\phi(a)$, von Mises functions ($\kappa = 2.$) arranged evenly on $[0, 2\pi]$. The policy iteration was run for 500 cycles, and in each cycle an episode of 500 steps was collected according to the greedy policy. The visited latent, inferred of observed state sequences were used to update the corresponding SFs to re-evaluate the policy. To facilitate faster learning, only episodes with positive returns were used to update the SFs.

# References

Arthur Gretton, Karsten M. Borgwardt, Malte J. Rasch, Bernhard Schölkopf, and Alexander Smola. A Kernel Two-Sample Test. *Journal of Machine Learning Research*, 13:723–773, March 2012. URL `http://jmlr.csail.mit.edu/papers/v13/gretton12a.html`.