[Reviews · NeurIPS 2019]

Reviewer 1



The paper is well-written and easy to follow. To my best knowledge, the work is novel and provide an extension to the successor representation. If data available, it would be interesting to see how similar or dissimilar human/animal behave to the proposed model, especially to examine the new point of view of the hippocampal function. The choice of experimental setting can be briefly explained for the readers not very familiar with relevant literature. For example, given that we interpret the results by looking at whether it learns the barrier, it would be good to explain why the barrier is curial to the experiment/model evaluation. _____________________ After authors response: The authors promised more discussion on neuroscience and comparison with earlier models, which would be valuable for this paper. So I am happy to increase my score by 1 point. Please also make some efforts to make the paper more readable by a boarder audience, as suggested by other reviewers and myself.

Reviewer 2



I don't have any major technical criticisms of the paper. However, I didn't feel that the experimental results really highlighted the advantages of this approach. Specifically, the authors never compare against any other method for solving POMDPs. I think this is necssary for making a compelling case for this method. Is it more sample efficient, more computationally efficient, more flexible? The links to neuroscience are very intriguing to me (as a neuroscientist), though I fully understand that the authors don't really have space to explore them. Nonetheless, one possibility might be to focus more on these applications instead of the toy experimental results. Then the impetus for demonstrating computational superiority over other methods is replaced with an impetus for demonstrating that this model can explain aspects of neural computation (e.g., in the hippocampus) that alternative models cannot. Minor: p. 3: "set features" -> "set of features" There seems to be inconsistent indexing of various quantities (e.g., M, V, T) by the policy pi. -------------- Comments after rebuttal: I was already quite enthusiastic about this paper. I don't think it's necessary to increase my rating. I'm happy that the authors are interested in providing more neuroscience context for their ideas.

Reviewer 3



The paper present a POMDP continuous approximation scheme for the RL successor representation, relevant when states are not known but must be inferred, allowing for a realistic learning algorithm in-between model-free and model-based RL. I am not aware of anyone having attempted something similar, clearly a very original submission. This is a dense paper (in a good way) with a lot of vey interesting results. The density meant that some of the results were a little hard to follow, but I do not doubt the importance of these results. This should be of relevance to a large proportion of NeurIPS attendees. After author response: I still think some more details on simulations would have beneficial, and I am not sure how much more the authors' response really adds, but I am still excited about this paper.

[Author Response · NeurIPS 2019]

# Response to reviewers

We would like to thank the reviewers for their positive feedback and suggestions.

In the final version of the paper we plan to provide additional explanation for the figures, as reviewer 1 and 2 have suggested.

As our primary focus is on biological plausibility, we will expand our discussion on links to existing hippocampal literature, as well as earlier biological reinforcement learning models that considered partial observability in much more restrictive settings (with discrete variables, exact inference).

In particular, distributional SRs provide a link between various intriguing and seemingly disparate experimental observations. The link between hippocampal physiology and (non-distributional) SRs has been explored previously (e.g., Stachenfeld 2017) providing an interpretation for phenomena such as "splitter" cells, which show spatial tuning that depends on the whole trajectory (i.e. policy) traversed by the animal not just on its current position (Grieves et al., 2016). However, as discussed in the paper, relevant states for a given reinforcement learning problem (in this case states over which the SR should be learned) can not be assumed to be directly available to the agent but must be inferred from observations. The hypothesis that hippocampal place cell activity encodes *inferred* location, with its concomitant uncertainty, has also been linked to experimental data (Madl et al., 2014). Thus, our approach connects these two separate threads in the literature and thereby encompasses both groups of experimental results.

Furthermore, the framework helps to link simulation of an internal model to learning. Acquisition of the inference model in our framework requires simulating experience (sleep samples) from the agent's current model of the environment which are then used to update the recognition model. The sleep samples reflect the agent's knowledge of the environmental dynamics but they don't necessarily correspond to a previously experienced trajectory exactly. This is reminiscent of hippocampal replays which do not just recapitulate previous experience, but often to correspond to novel trajectories not previously experienced by the animal (Gupta et al., 2010; Olafsdottir et al., 2015; Stella, 2019). Relatedly, Liu et al. (2019) recently observed that replay events in humans reflect abstract structural knowledge of a learned task. Our model suggests a novel functional interpretation of these replayed trajectories, namely, that they may play an important role in *learning to infer* relevant latent states from observations. This accords with the observation that experimental interference with replay events impedes learning in contexts where optimal actions depend on history-based inference (Jadhav et al., 2012).

[Meta-Review · NeurIPS 2019]

This work proposes a neurally plausible approach to reinforcement learning in partially-observed MDPs based on distributional successor features. The approach allows for efficient value function computation as demonstrated empirically. The three expert reviewers were unanimous that this paper should be accepted, and I see no reason to contradict their opinions.